# Flow Cytometric Analysis of Bone Marrow Particle Cells for Measuring Minimal Residual Disease in Multiple Myeloma

**DOI:** 10.3390/cancers14194937

**Published:** 2022-10-08

**Authors:** Duanfeng Jiang, Yanan Zhang, Shiming Tan, Jing Liu, Xin Li, Congming Zhang

**Affiliations:** 1Department of Hematology, The Third Xiangya Hospital, Central South University, Changsha 410013, China; 2Department of Hematology, Second Affiliated Hospital of Hainan Medical College, Haikou 570311, China

**Keywords:** minimal residual disease (MRD), flow cytometry, bone marrow particle cells (BMPLs), multiple myeloma (MM)

## Abstract

**Simple Summary:**

The hemodilution of bone marrow aspirates is the main factor affecting the reliability of the determination of the minimal residual disease (MRD) resulting in multiple myeloma (MM). A bone marrow particle cell (BMPL) enrichment assay may be applied to overcome the problems caused by the hemodilution of bone marrow aspirates and to improve the rate of myeloma cell detection. Assessment of BMPL samples could decrease the false-negative rate of the MRD assessed with routine bone marrow aspirates. This study validated that the BMPL was superior to the routine bone marrow aspirate samples in the MM and MRD analysis.

**Abstract:**

Minimal residual disease (MRD) is one of the most relevant prognostic factors in patients with multiple myeloma (MM). However, the hemodilution of bone marrow (BM) aspirates, the most common preanalytical problem, is known to affect MRD detection. In the present study, we analyzed a preanalytical method for routine BM aspirates and a bone marrow particle cell (BMPL) enrichment assay and validated it as a reliable preanalytical method for flow cytometric MRD determination. A total of 120 BM samples were taken from 103 MM patients consecutively recruited; 77 BM samples had BMPL enrichment analysis and 99 BM samples were routinely analyzed. Then, the two different samples from patients with MM were sent for MRD detection using an eight-color flow cytometry. Our data showed that assessment of the BMPL enrichment samples attenuated the overestimation of MRD-negative assessed in the routine BM samples, which was mainly caused by hemodilution. In conclusion, the BMPL enrichment assay is a functional and practical preanalytical method for flow cytometric MRD analysis.

## 1. Introduction

Multiple myeloma (MM) is a plasma cell malignancy that remains incurable despite great improvements in treatments during the past decades [1,2]. The treatment of MM has benefited from new specific and sensitive means for disease prediction and management [3,4]. The evaluation of the minimal residual disease (MRD) after treatment is one of these means. The more sensitive the technique used and the lower the MRD level that is obtained with the technique, the longer the progression-free survival (PFS) is predicted to be [5]. The methods for measuring MRD in the bone marrow (BM), such as next-generation flow (NGF) cytometry and next-generation sequencing (NGS), have a sensitivity in the range of 10^−5^ to 10^−6^ cells [6]. These techniques can be combined with functional imaging to evaluate the MRD outside the BM [7,8]. The MRD could be used as an alternative endpoint for clinical trials to accelerate the approval of new drugs for MM [7]. Most importantly, the MRD status could be a predictor for making treatment decisions and clarifying the timing of an intervention [5].

As current therapies achieve an unprecedented depth of response, more sensitive methods for response evaluation and MRD detection are urgently needed [9]. Multiple studies have consistently shown that the PFS in MRD-negative patients is better when using more sensitive flow cytometry to detect tumor cells [10,11]. Most patients with complete remission (CR) eventually relapse; therefore, more sensitive methods are needed to detect and quantify the MRD. Multiparametric flow cytometry (MFC) [12] is essential for quantifying the MRD in MM patients [6]. However, hemodilution of the sample (mixed peripheral blood (PB) in the BM) may lead to an underestimation of the disease burden [13]. Hemodilution of the BM aspirate sample is the most common preanalytical problem [14]. Traditionally, the first “pull” of a marrow aspiration is not sent for MRD detection using flow cytometry, which can lead to significant hemodilution and underestimation of the MRD [14,15]. It is of the utmost importance to assess the quality of the samples prior to flow cytometry analysis. Here, a simple method to optimize routine BM aspirates, bone marrow particle cell (BMPL) enrichment analysis, was used as a preanalytical solution to improve the BM quality [16].

In the current study, we performed a validation study of the sample preparation technique [16], the preanalytical method for routine BM aspirates, and its implementation in flow cytometry assessment. An eight-color “cell subset distribution” tube was used to characterize and quantify samples from patients with MM sent for assessment of MRD. The relationship between the MRD and clinical benefit were confirmed in each disease context. It was validated that the BMPL enrichment assay minimized hemodilution in the traditional BM samples. Moreover, this study assessed the frequency and degree of hemodilution in a cohort of patients with MM. 

## 2. Materials and Methods

### 2.1. Patient Samples and Characteristics

Bone marrow (BM) samples and clinical data were obtained from patients who were diagnosed with MM [2] or monoclonal gammopathy of undetermined significance/smoldering multiple myeloma (MGUS/SMM) [17] between 31 March 2019 and 1 April 2021. A total of 120 BM samples were taken from 103 consecutively recruited patients (some were measured more than once, e.g., before and after autologous hematopoietic stem cells of the same patient) with MM (from MGUS to MM), who underwent routine BM aspiration. Among the 120 samples, 56 samples from the same patients were divided into two parts (conventional BM and BMPL enrichment samples). Finally, a total of 77 BM samples underwent BMPL enrichment analysis and 99 BM samples were routinely analyzed. The samples of patients diagnosed with MGUS/SMM (*n* = 9) or newly diagnosed MM (NDMM) (*n* = 16) were analyzed at diagnosis before treatment. The MM patient samples (*n* = 78) were analyzed after 4–6 cycles of standard chemotherapy or 3–12 months post autologous stem cell transplantation (auto-SCT) for MRD detection. Samples were analyzed within 6 h of aspiration. Informed consent was obtained according to the Declaration of Helsinki. This study was reviewed and approved by The Third Xiangya Hospital of Central South University (protocol code: 2022-76) (09/2022).

### 2.2. Hemodilution Assessment of Bone Marrow Aspirates

To assess the hemodilution, we used several proposed methods: (a) The Holdrinet index (HI) [18], the gold standard for quantifying hemodilution, whose formula is: HI = (BM erythrocytes × PB leukocytes)/(BM leukocytes × PB erythrocytes); HI > 0.5 indicates hemodilution. (b) The ratio of immature granulocytes and neutrophils (IGRA/N): a ratio of ≥1.2 indicates a pure BM sample, while a ratio of <0.5 indicates a high degree of hemodilution [19]. (c) The peripheral blood contamination index (PBCI) [20]: PBCI = –3.052 + 6.5 × (mature neutrophils/granulocytes) − 0.609 × (%CD34) − 2.008 × (% plasma cells); a value above 1.2 indicates significant contamination of the BM sample with PB. Pearson correlation analysis was performed for the correlation of the Holdrinet index with the IGRA/N ratio, PBCI, and %CD34 cells.

### 2.3. Enrichment of Bone Marrow Particle Cells (BMPLs)

Approximately 3–5 mL of the BM aspiration was enriched according to the sample preparation and procedure described in a previous study [16]. The details of the method are shown in the Appendix A.

### 2.4. Flow Cytometric MRD Analysis

Flow cytometric MRD assays were performed on the BM aspirates in our laboratory. The BM processing was completed within 24 h of collection according to the EuroFlow guidelines [12,21]. The BM samples and BMPL enriched samples were lysed and stained with the cell-surface antibodies, CD19, CD27, CD38, CD45, CD56, and CD138 [22], and the cytoplasmic antibodies, kappa (κ) and lambda (λ) (Appendix A) included in the eight-color panels [21]. For all analyses, we acquired at least 5 × 10^5^ events for each sample. Flow cytometric MRD-negative (MRD^−^) was defined as the detection of <50 abnormal plasma cells (aPCs) among the collected nucleated cells [21,22,23]. The aPCs were typically CD138^+^CD38^+^CD27^−/dim^CD56^+/−^CD45^−^CD19^−^. The normal plasma cells (nPCs) were typically CD138^+^CD38^+^CD45^+^CD27^+^CD56^−/dim^ CD19^+/−^ and polyclonal for κ and λ. 

### 2.5. Statistical Analysis

The differences between the continuous variables were determined using the unpaired *t* test or the Mann–Whitney U test. Paired *t* test was used to analyze the difference between BM samples and BMPL samples from the same patients. The overall survival (OS) and progression-free survival (PFS) were calculated and plotted with Kaplan–Meier curves and compared using the log-rank test. The statistical significance was defined as *p* < 0.05. Statistical analyses were performed using GraphPad Prism 7.0.

## 3. Results

### 3.1. Clinical and Laboratory Characteristics of MM Patients

To evaluate the distinct sample preparation protocols (the BMPL enriched samples against the conventional BM samples) and to validate the BMPL samples used for eight-color flow cytometric MRD, a total of 120 BM samples from 103 patients were consecutively recruited. The characteristics of the two groups of patients according to the BM samples with (*n* = 77) or without (*n* = 99) particle cell enrichment in terms of age, sex, and disease characteristics are shown in Table 1. Patients’ characteristics in the BM and BMPL cohorts were comparable. The response to treatment was assessed according to the International Myeloma Working Group criteria [6].

### 3.2. Hemodilution Assessment in Patients with Multiple Myeloma

The burden of MRD may be the best prognostic indicator obtained from BM aspiration [14]. However, the MRD samples are often obtained last or in a non-optimal sequence which can result in marked hemodilution and underestimation of the MRD [15]. To assess the hemodilution in our center, we used several proposed methods [18,19,20] with the 120 BM samples. As shown in Table 2, the Holdrinet index (HI) was >0.5 in 32/90 samples (35%), the IGRA/N was <0.5 in 19/96 samples (20%), and the PBCI was >1.2 in 8/56 samples (14%). These data showed that there was a high possibility of hemodilution in the conventional BM samples used for flow cytometric MRD detection. In addition, the correlation between the HI and the IGRA/N ratio was −0.65, and that between the HI and PBCI was 0.58. The median level of the CD34^+^ myeloblasts in the BM was 0.37%.

### 3.3. MRD Detection by Flow Cytometry

To overcome the hemodilution in the traditional BM aspirates, a novel method, BMPL enrichment analysis, has been used for diagnosis of difficult and complicated MM cases [16]. In this study, the BMPL enrichment method was further validated in the flow cytometry protocol for detection of the MRD among MM patients [23]. A total of 25 patients at new diagnosis and 78 patients after treatment were assessed with BM samples or BMPL samples (Table 1). As shown in Figure 1A and B, significantly higher (*p* < 0.001) numbers of aPCs were observed in the NDMM patients (*n* = 16) than in those with MGUS/SMM (*n* = 9), indicating that increasing aPCs correlated with patients’ disease status. In addition, it has been previously described that an increase in aPCs was observed in the BM of MM patients, when compared with that of the premalignant disease [24]. Notably, consistent conclusions could be reached using the preanalytical enrichment assay (Figure 1B), supporting the reliability of the BMPL enrichment method for MRD measurement.

Age, sex, and MM characteristics were similar in the conventional BM sample (*n* = 74) and BMPL sample (*n* = 60) groups (Table 1). In the entire MRD cohort (*n* = 78), 56 patient BM aspirates were divided into two parts—one was processed with the BMPL enrichment assay, the other had no preanalytical processing. The samples were analyzed for MRD at a limit of detection (LOD) of 10^−4^ [11,25] which was achieved in 66 (89%) conventional BM samples (Figure 1C) and in 57 (95%) enriched BMPL samples (Figure 1D). According to the MRD cutoff (<10^−4^), 27% (*n* = 16) reached MRD^−^ and 73% (*n* = 44) remained MRD^+^ in the BMPL samples (Appendix A). These results were consistent with previous studies using the flow cytometry approach for the assessment of MRD status [26]. By comparison, 40% (*n* = 30) reached MRD^−^ and 60% (*n* = 44) remained MRD^+^ in the BM samples (Appendix A), indicating that the proportion of MM patients evaluated as MRD^−^ with the traditional BM samples was much higher than with the BMPL samples. Furthermore, the number of aPCs was higher (unpaired samples: *p* = 0.0024, Figure 1E; paired samples: *p* < 0.001, Figure 1F) in the BMPL samples than in the BM samples. As expected, the number of aPCs was evidently higher (*p* < 0.001) in MRD^+^ cases than in MRD^−^ patients measured either in the BM samples (Figure 2A) or in the BMPL samples (Figure 2B).

In the BM samples, 90% of the patients in the MRD^−^ cohort were in complete remission (CR) or very good partial remission (VGPR) vs. 84% in the MRD^+^ group; while in the BMPL samples, 100% of the patients in the MRD^−^ cohort were in CR or VGPR vs. 84% in the MRD^+^ group, indicating that the MRD^−^ rates were underestimated in the assessment of the MRD levels in the CR patients. In addition, 14% of the patients in the MRD^−^ cohort progressed vs. 20% in the MRD^+^ group assessed with the BM samples; whereas 6% of the patients in the MRD^−^ cohort progressed vs. 20% in the MRD^+^ group assessed with the BMPL samples. Notably, the two different samples were equally bright and reliable for aPC and nPC detection, despite the varied phenotypes between MM patients. Moreover, in three cases (23%) with uncertain phenotype, the numbers of aPCs were faint by flow cytometry though high via histology. The BMPL enrichment assay was more precise and reliable for aPC detection (Appendix A). These data indicate that the use of the BMPL enrichment assay seems to weaken the underestimation of disease diagnosis or overestimation of MRD^−^ caused by hemodilution.

Since MRD^−^ is correlated with superior PFS and OS [27], similar results were observed in this study. The median observation time was 24.7 months (range: 5.4–39). The MM patients with the conventional BM samples showed no significant differences between the MRD^+^ group and the MRD^−^ group in OS (*p* = 0.071) (Figure 2C). However, the MRD^−^ group showed a longer OS than the MRD^+^ group when assessed with the BMPL samples (*p* = 0.028) (Figure 2D). The median PFS in the MRD^+^ cases and MRD^−^ patients were not reached in the conventional BM samples (*p* = 0.29) (Figure 2E). Consistently, the median PFS in the MRD^+^ group and MRD^−^ group were not reached in BMPL samples (*p* = 0.10) (Figure 2F). These data suggested that both the BM and BMPL samples were reliable for MRD assessment, and it seems that the MRD evaluated by the BMPL samples played a better role in prognosis indication. 

In addition to the aPC analysis, another focus was to determine the nPCs. The numbers of nPCs were evidently higher (*p* = 0.023) in the BMPL samples than in the traditional BM samples from the MRD cohort (Figure 3A), suggesting a remarkable enrichment effect on the BM aspirates. To explore the role of the nPCs in more detail, the MRD samples were divided into two groups: after standard chemotherapy (CTx) (BM samples: *n* = 30; BMPL samples: *n* = 26) and after auto-SCT (BM samples: *n* = 44; BMPL samples: *n* = 34). As shown in Figure 3B, the BMPL samples from patients after auto-SCT had higher (*p* = 0.0012) numbers of nPCs compared to the BM samples from patients after auto-SCT, suggesting an enrichment function of the BMPL samples. However, there was no significant difference (*p* = 0.69) in the numbers of nPCs between the BMPL samples and the BM samples from patients after CTx. It is possible that the numbers of nPCs in patients after CTx had a wider variation than that in patients after auto-SCT. Of note, the nPC numbers were higher (BM: *p* = 0.0067; BMPL: *p* = 0.0025) in patients after auto-SCT than in patients after CTx. The increase in the number of nPCs after auto-SCT may indicate that the BM state of MM patients returns to normal. These findings were similar to previous studies [24,26]; however, the nPC numbers appeared markedly higher in the MRD^−^ group than in the MRD^+^ groups, either assessed with the BM samples (*p* = 0.041) (Figure 3C) or with the BMPL samples (*p* < 0.001) (Figure 3D). The numbers of nPCs in the MRD^−^ group were significantly higher possibly because of a closer status to normal BM, as healthy individuals had higher nPC numbers than MM patients at different disease statuses [24,26].

## 4. Discussion

Preanalytical factors, such as the BM sample quality, have a strong influence on the determination of the MRD. The most common preanalytical problem of the BM aspirate sample is hemodilution [15]. Traditional BM aspiration sent for flow cytometry is often not obtained from the first “pull” or is obtained in an inconsistent sequence, which may lead to underestimation of the MRD [14]. In this study, we validated a new method called bone marrow particle cell (BMPL) enrichment analysis for MRD, which has been developed for diagnosis of rare MM cases [16]. 

We first evaluated the hemodilution of the routine BM aspirates sent for flow cytometric MRD analysis. A high incidence of hemodilution was observed in the BM aspirate samples, and the IGRA/N ratio may help to distinguish the patients whose MRD could be underestimated. Based on these results, we applied the BMPL enrichment analysis for flow cytometric MRD and aimed to find a reliable solution to hemodilution. Acquiring large numbers of cells appears to be a prerequisite for obtaining MRD data with good sensitivity. The new preanalytical protocol, BMPL enrichment, allowed acquisition of high cell numbers [16]. BMPL enrichment analysis is simple, easy to perform, and could be applied for almost all conventional BM aspirates [16]. In this study, BMPL enrichment analysis increased the sensitivity from 89% of patients’ samples to 95% of patients’ samples. Our data suggested that routine BM aspiration is feasible after enrichment analysis, although high quality BM samples remain important [14,22]. 

Our MRD determination was at month 3–12 after standard treatment or auto-SCT, according to the routine BM assessment procedure in our centers. In the BMPL enriched samples, 100% of patients in the MRD^−^ group were in CR or VGPR; while in the conventional BM samples, 90% of patients in the MRD^−^ cohort were in CR or VGPR, suggesting that the assessment of routine BM samples for the MRD levels may overestimate MRD^−^ rates. Our MRD^−^ rate of 27% assessed with the BMPL samples was comparable to that of other analyses [21,26] but somewhat lower than the 40% assessed with the BM samples, at least in part due to the hemodilution resulting in overestimation of the MRD^−^ rates. Furthermore, only 6% of the patients progressed in the MRD^−^ group assessed with the BMPL samples, whereas 14% of patients did so in the MRD^−^ cohort assessed with the BM samples, indicating some MRD^−^ patients may be overestimated. These findings were similar to previous reports [26,28,29]; this concordance is probably owing to the higher sensitivity of BMPL samples, resulting from the larger number of cells analyzed. However, fewer patients in the MRD cohort were in CR, mainly because of the shorter time between the start of standard treatment and MRD detection. In general, our data suggested that both samples were available for MRD assessment, and MRD evaluated by the BMPL samples seemed more accurate and reliable. 

Our data showed that the results concerning the MRD-negative and improved PFS and OS, no matter whether they were analyzed with the conventional BM samples or the BMPL samples, were similar to those in large clinical studies [27,30]. We emphasized the practicability of the application of the BMPL enrichment assays in MRD detection by flow cytometry. Additionally, we explored the significance of the nPC numbers in the MRD cohort. The numbers of nPCs were higher in the MRD^−^ groups than in the MRD^+^ groups, which was contrary to previous studies [26], and is worthy of further study. It is consistent with the literature that higher nPC numbers after auto-SCT may suggest a reversion to normal BM status. According to the literature [26] that evidences higher nPCs in healthy individuals than in plasma cell disorder (PCD) patients and higher nPCs in MGUS/SMM than in MM patients, we suppose the nPC numbers may reflect the status of normal BM and may have the potential to be an indicator of superior prognosis. 

In summary, our study addressed the urgent need for reliable and sensitive flow cytometry preanalytical methods to assess therapeutic efficacy, to improve on the conventional BM samples. Our study validated that the BMPL enrichment assay was a functional and practical preanalytical method for flow cytometric MRD detection. However, several limitations need to be acknowledged. The time between treatment and MRD detection was too short and the number of samples was not large. Further studies will focus on the comparison of samples assessed with both methods, as well as the mechanically disaggregated BM trephine biopsy [31] samples for flow cytometric analysis.

## Figures and Tables

**Figure 1 cancers-14-04937-f001:**
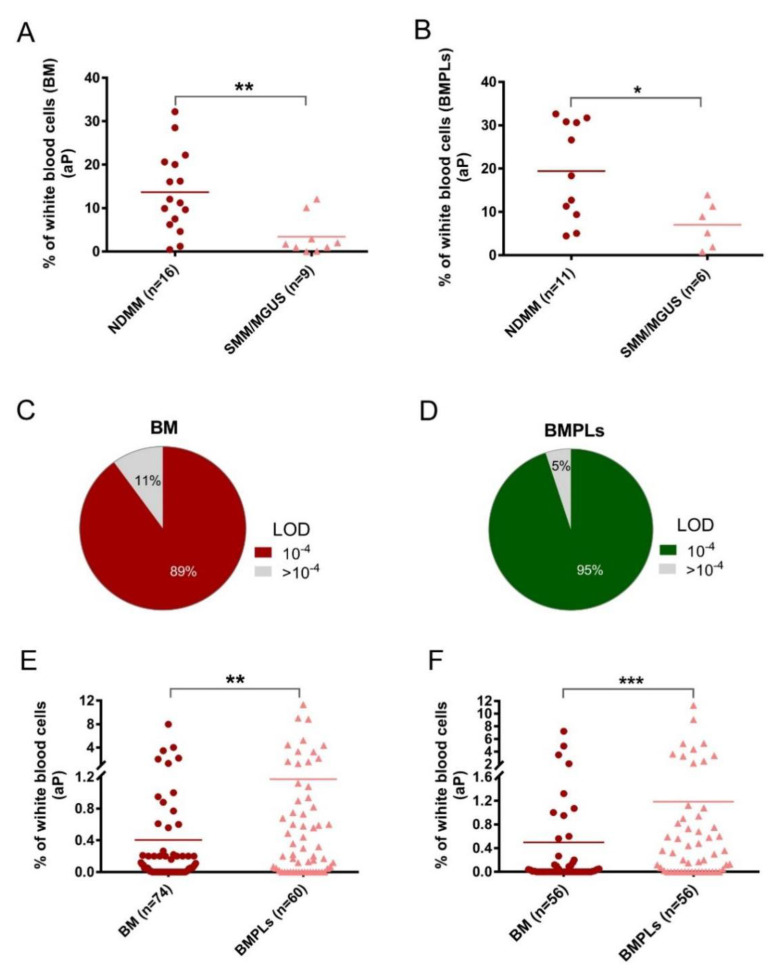
(**A**,**B**) Two groups of patients’ samples at initial diagnosis, newly diagnosed multiple myeloma (NDMM) and smoldering multiple myeloma (SMM)/monoclonal gammopathy of undetermined significance (MGUS), were assessed with conventional bone marrow (BM) or enriched bone marrow particle cells (BMPLs). The percentage of aberrant plasma cells (aPCs) was higher (BM: ** *p* = 0.0053; BMPLs: * *p* = 0.016) in the NDMM samples (BM: *n* = 16; BMPLs: *n* = 11) than in the SMM/MGUS samples (BM: *n* = 9; BMPLs: *n* = 9). (**C**,**D**) In 89% of the BM samples (*n* = 74) and in 95% of the BMPL samples (*n* = 60) for minimal residual disease (MRD), the limit of detection (LOD) and a sensitivity of 10^−4^ was reached. (**E****,F**) The percentages of aPCs in the MRD samples assessed with BMPL (*n* = 60) were higher (**E****:** unpaired samples, ** *p* = 0.0024; **F:** paired samples, *** *p* < 0.001) in the BMPL samples than in the BM samples) than those in the samples assessed with BM (*n* = 74). (F) The graphs show the mean (**A**,**B**,**E**,**F**), Mann–Whitney U test or paired *t* test.

**Figure 2 cancers-14-04937-f002:**
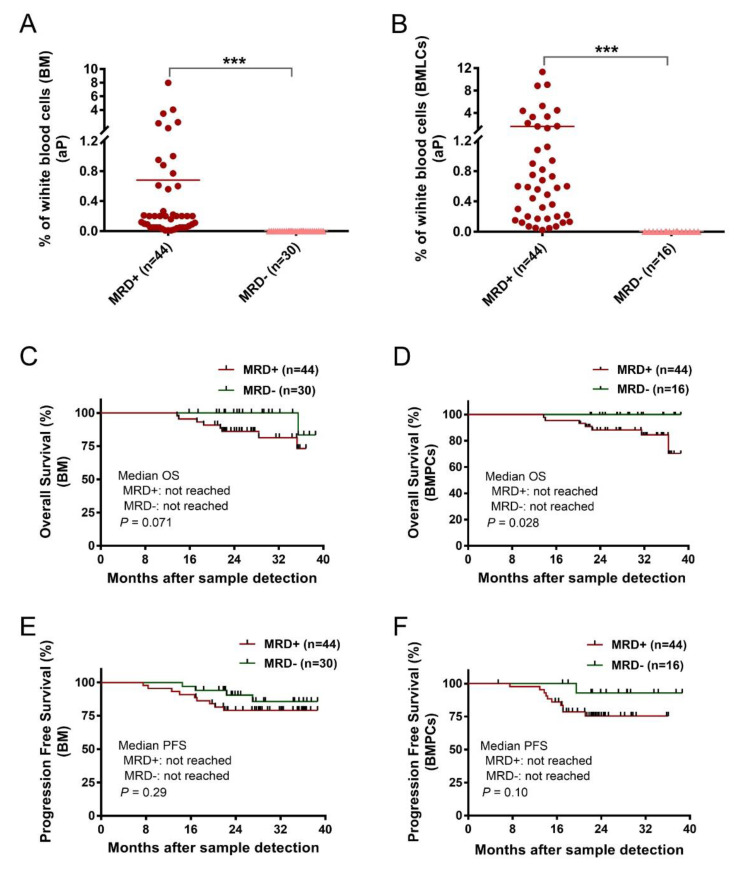
(**A**,**B**) Two groups of patients’ samples after treatment, MRD^+^ (≥10^−4^) and MRD^−^ (<10^−4^), were assessed with conventional bone marrow (BM) samples or enriched bone marrow particle cell (BMPL) samples. The percentages of aberrant plasma cells (aPCs) in the MRD^+^ samples (BM: *n* = 44; BMPL: *n* = 44) were significantly higher (BM: *** *p* < 0.001; BMPL: *** *p* < 0.001) than those in the MRD^−^ samples (BM: *n* = 30; BMPL: *n* = 16). (**C**,**D**) BM (*n* = 60) or BMPL (*n* = 74) samples from patients after therapy were divided into MRD^+^ (≥0.001% aPC) and MRD^−^ (<0.001% aPC). Kaplan–Meier was used to estimate the overall survival (OS); no differences were observed between the MRD^−^ group and the MRD^+^ group in the BM samples (*p* = 0.071) (**C**); however, the MRD^−^ patients had a longer OS than the MRD^+^ cohort in the BMPL samples (*p* = 0.028). (**D**–**F**) Progression-free survival (PFS) was determined for the MRD^−^ and MRD^+^ cohort in the BM samples (*p* = 0.297) and in the BMPL samples (*p* = 0.101). The graphs show the mean (**A**,**B**), Mann–Whitney U test (**A**,**B**), and the Kaplan–Meier method (**C**–**F**).

**Figure 3 cancers-14-04937-f003:**
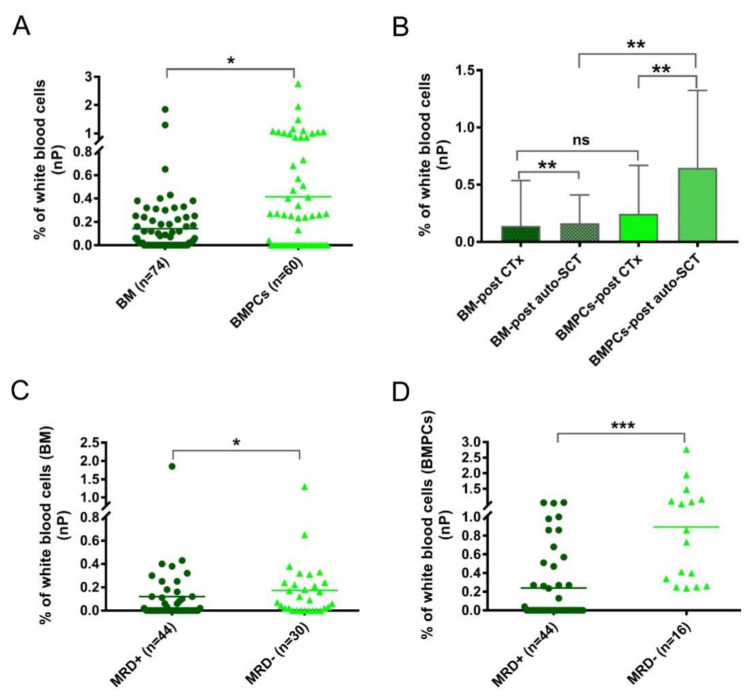
(**A**) The percentages of normal plasma cells (nPC) were determined with conventional bone marrow (BM) or enriched bone marrow particle cell (BMPL) samples from patients after treatment. The percentages of nPCs were higher (* *p* = 0.023) in the BM samples (*n* = 74) than in the BMPL samples (*n* = 60). (**B**) Comparison of the nPC numbers in patients with different treatment regimes and different samples (BM and BMPL). The percentages of nPCs were higher (BM: ** *p* = 0.0067; BMPL: ** *p* = 0.0025) in patients after autologous stem cell transplantation (auto-SCT) (BM: *n* = 44; BMPL: *n* = 34) than in patients after treatment with standard chemotherapy (CTx) (BM: *n* = 30; BMPL: *n* = 26). The BMPL samples from patients after auto-SCT had higher (** *p* = 0.0012) percentages of nPCs compared to the BM samples from patients after auto-SCT. No significant differences were observed between the BM samples and BMPL samples from patients after treatment with CTx. (**C**,**D**) Patients’ samples after treatment were divided into MRD^+^ (≥0.001% aPC) and MRD^−^ (<0.001% aPC). The percentages of nPCs were higher (BM: * *p* = 0.041; BMPL: *** *p* < 0.0001) in the MRD^−^ samples than in the MRD^+^ samples. The graphs show the mean (**A**,**C**,**D**) or mean ± standard deviation (**B**), Mann–Whitney U test (**A**–**D**).

**Table 1 cancers-14-04937-t001:** Characteristics of the patients in BM or BMPLs group of total cohort or MRD cohort.

Patient’s Parameters	Total Cohort (*n* = 103) ^a^		*p* Value	MRD Cohort (*n* = 78) ^a^		*p* Value
	BM (*n* = 99)	BMPLs (*n* = 77)		BM (*n* = 74)	BMPLs (*n* = 60)	
Disease setting (%)			0.98			0.74
MM	90 (91)	71 (92)		-	-	
NDMM	16 (16)	11 (14)		-	-	
Post CTx	30 (30)	26 (34)		30 (41)	26 (43)	
Post auto-SCT	44 (45)	34 (44)		44 (59)	34 (57)	
SMM/MGUS	9 (9)	6 (8)		-	-	
Median age (range)	59 (34–79)	58 (34–78)	0.70	58 (34–78)	59 (34–78)	0.82
Sex (%)			0.35			0.35
Female	42 (42)	27 (36)		33 (45)	22 (37)	
Male	57 (58)	49 (64)		41 (55)	38 (63)	
Durie-Salmon (%)			0.64			0.92
I	12 (12)	7 (9)		3 (4)	3 (5)	
II	22 (22)	21 (28)		17 (23)	15 (25)	
III	65 (66)	48 (63)		54 (73)	42 (70)	
R-ISS (%)			0.87			0.91
I	13 (13)	8 (11)		4 (5)	3 (5)	
II	44 (45)	35 (46)		33 (45)	29 (48)	
III	42 (42)	33 (43)		37 (50)	28 (47)	
MM type (%)			0.88			0.92
IgG	58 (59)	45 (59)		40 (54)	32 (53)	
IgA	24 (24)	21 (28)		20 (27)	19 (32)	
IgD	2 (2)	2 (3)		2 (3)	2 (3)	
Light chain only	14 (14)	7 (9)		11 (15)	6 (10)	
Biclonal	1 (1)	1 (1)		1 (1)	1 (2)	
Light chain (%)			0.84			0.78
Kappa	46 (46)	32 (42)		34 (46)	24 (40)	
Lambda	52 (53)	43 (57)		39 (53)	35 (58)	
Biclonal	1 (1)	1 (1)		1 (1)	1 (2)	
MM Progression (%)			0.59			0.89
Yes	14 (14)	13 (17)		13 (18)	10 (17)	
No	85 (86)	63 (83)		61 (82)	50 (83)	
Vital status (%)			0.97			0.91
Dead	4 (4)	3 (4)		4 (5)	3 (5)	
Alive	95 (96)	73 (96)		70 (95)	57 (95)	
% PC of BM (range)	-	-		1(0–21)	2 (0–14)	0.016
Cytogenetics b (%)						0.94
High-risk	-	-		22 (30)	18 30)	
Intermediate-risk	-	-		9 (12)	8 (13)	
Standard-risk	-	-		33 (45)	28 (47)	
Missing	-	-		10 (13)	6 (10)	
Remission ^c^ (%)						0.98
CR	-	-		17 (23)	14 (23)	
VGPR	-	-		47 (64)	39 (65)	
PR	-	-		8 (11)	6 (10)	
SD	-	-		2 (2)	1 (2)	

^a^ The number of two groups together is higher than that of the whole cohort because some patients were measured more than once during the period of sampling. ^b^ High risk defined as: del 17p, t(14;16), t(14;20), +1q, del 1p; Intermediate risk defined as: t(4;14), del 13q, hyperdiploidy; Standard risk defined as: t(11;14), t(6;14), del 14q (IgH), others. ^c^ According to International Myeloma Working Group criteria. Abbreviation: BM: bone marrow; BMPLs: bone marrow particle cells; MRD: minimal residual disease; MM: multiple myeloma; NDMM: newly diagnosed multiple myeloma; post-CTx: after standard chemotherapy; post-ASCT: after autologous stem cell transplantation; SMM: smoldering multiple myeloma; MGUS: monoclonal gammopathy of undetermined significance; PC: plasma cells; CR: complete response; VGPR: very good partial response; PR: partial response; and SD: stable disease.

**Table 2 cancers-14-04937-t002:** Bone marrow hemodilution in patients with multiple myeloma after treatment.

	*n*	Median	Interquartile Range	Correlation with Holdrinet Index	*p* Value
Holdrinet index ^a^	90	0.41	0.21–0.62	-	
IGRA/N ratio ^b^	96	1.14	0.73–1.94	−0.65	<0.01
PBCI ^c^	56	−1.02	−1.92 to 0.23	0.58	0.023
%CD34 cells	60	0.37%	0.06–1.68%	−0.37	0.041

^a^ Holdrinet index < 0.3 in 34/90 (38%), between 0.3 and 0.5 in 24 (27%), and >0.5 in 32 (35%). ^b^ IGRA/N ≥ 1.2 in 46/96 (48%), between 0.5 and 1.2 in 31 (32%), and <0.5 in 19 (20%). ^c^ PBCI > 1.2 in 8/56 (14%). Abbreviations: *n*, number; IGRA/N, immature granulocytes/neutrophils ratio; and PBCI, peripheral blood contamination index. (Pearson r = −0.41, *p* = 0.0021, with F test).

## Data Availability

The data presented in this study are available in this article (and Appendix A).

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
