# Peer review of "Flow Cytometric Analysis of Bone Marrow Particle Cells for Measuring Minimal Residual Disease in Multiple Myeloma"

_cancers, 2022, doi:10.3390/cancers14194937_

Round 1

Reviewer 1 Report

In this article, Jiang et al reported a method avoiding hemodilution of bone marrow aspirates samples to detect precise minimal residual disese of multiple myeloma. 

I totally agree to the authors comments that there is urgent need for reliable and sensitive methods for MRD detection.

While the method is interesting to some extent, it seems to have little clinical impact for the complexity of the method. The clinical superiority (i.e. clear difference of survival) of this method needs to be more clearly demonstrated in order to apply it to actual clinical practice.

It is difficult to understand the method only by text, so it would be better if it is represented by a diagram.

In the figure 2A and 2B, how about compare abnormal plasma cells in BM by conventional method and BMPL?

In the figure 2, there are confusing there is confusion in the use of terminology BMLCs, BMPCs, BMPL. It is confusing. 

This article may be suitable for other journals dealing with methodology. 

Reviewer 2 Report

Bone marrow particle cells (BMPLs) enrichment assay which is a pre-analytical method for routine bone marrow aspirates.
In this interesting and novel study, the authors aimed to validate it as a reliable pre-analytical method for flow cytometric minimal residual disease (MRD) determination in a cohort of patients with multiple myeloma. Their data  shows that hemodilution could cause overestimation of MRD-negative populations and BMPLs enrichment could reduce this.
The abstract and summary are lucid and conclusions are well-thought out and references are comprehensive.

Minor comments:
1. A short section for sample preparation and staining procedure could be included in the methods or provided as supplemental data.
2. One suggestion is to fit Table 1 into a single page as it would be visually unchallenging.
3. Language editing is needed. 

Reviewer 3 Report

In the paper “Flow cytometric analysis of bone marrow particle cells for measuring minimal residual disease in multiple myeloma” Jiang et al., used multicolored flow cytometry analysis from two different samples (77 BM samples experienced BMPLs enrichment analysis and 99 BM samples were routinely analyzed) to characterize and quantify samples from patients with MM sent for assessment of MRD. The results conclude that BMPLs enrichment assay is a functional and practical pre-analytical method to detect MRD through Flow cytometry analysis. Overall, the study is well presented however, the below comments should be addressed.

Major comments

1.      In the methods section it is showing that “A total of 120 BM samples were taken from 103 consecutively recruited patients (some were measured more than once) with MM (from MGUS to MM), who underwent routine BM aspiration”. Authors should discuss how 120 BM samples were collected from 103 patients, if some were measured more than once what is the rationale to collect more than once?   

2.      In the methods section it is showing that “Among the 120 samples, 56 samples were divided into two parts (conventional BM and BMPLs enrichment samples)”. But in the abstract, they were showing 77 BM samples were used for BMPLs enrichment analysis and 99 BM samples were routinely analyzed, the n number in the abstract is different from the methods section, the authors should clarify about n number more specifically in the methods section.

3.      In the sentence BM samples ……… cytoplasmic antibodies kappa (κ) and lambda (λ) (Supplementary Table S1) included in 8-color panels (21), I could not find any Supplementary Table S1 in this manuscript, if this is from Ref. 21 it should be the supplementary table 2 from the Flores-Montero et al., Leukemia. 2017; 31, 2094–2103 (cited as 21).

4. The entire study was based on flow cytometry, surprisingly for MRD detection by flow cytometry, there no flow cytometry representative plots/results were shown for these panels (aPC were typically CD138+CD38+CD27–/dimCD56+/–CD45–CD19–. Normal plasma cells (nPC) were typically CD138+CD38+CD45+CD27+CD56–/dim CD19+/– and polyclonal for κ and λ) in the manuscript. It should be informative if the authors can show the flow gating strategy and the results by showing the flow profiles for MRD detection.